# MIND THE GAP
# WHEN CONDITIONING AMORTISED INFERENCE
# IN SEQUENTIAL LATENT-VARIABLE MODELS

**Justin Bayer**[*]     **Maximilian Soelch**[*]

**Atanas Mirchev**     **Baris Kayalibay**     **Patrick van der Smagt**

Machine Learning Research Lab, Volkswagen Group, Munich, Germany
`{bayerj,m.soelch}@argmax.ai`

## ABSTRACT

Amortised inference enables scalable learning of sequential latent-variable models (LVMs) with the evidence lower bound (ELBO). In this setting, variational posteriors are often only partially conditioned. While the true posteriors depend, e. g., on the entire sequence of observations, approximate posteriors are only informed by past observations. This mimics the Bayesian filter—a mixture of smoothing posteriors. Yet, we show that the ELBO objective forces partially-conditioned amortised posteriors to approximate products of smoothing posteriors instead. Consequently, the learned generative model is compromised. We demonstrate these theoretical findings in three scenarios: traffic flow, handwritten digits, and aerial vehicle dynamics. Using fully-conditioned approximate posteriors, performance improves in terms of generative modelling and multi-step prediction.

## 1 INTRODUCTION

Variational inference has paved the way towards learning deep latent-variable models (LVMs): maximising the evidence lower bound (ELBO) approximates maximum likelihood learning (Jordan et al., 1999; Beal, 2003; MacKay, 2003). An efficient variant is amortised variational inference where the approximate posteriors are represented by a deep neural network, the inference network (Hinton et al., 1995). It produces the parameters of the variational distribution for each observation by a single forward pass—in contrast to classical variational inference with a full optimisation process per sample. The framework of variational auto-encoders (Kingma & Welling, 2014; Rezende et al., 2014) adds the reparameterisation trick for low-variance gradient estimates of the inference network. Learning of deep generative models is hence both tractable and flexible, as all its parts can be represented by neural networks and fit by stochastic gradient descent.

The quality of the solution is largely determined by how well the true posterior is approximated: the gap between the ELBO and the log marginal likelihood is the KL divergence from the approximate to the true posterior. Recent works have proposed ways of closing this gap, suggesting tighter alternatives to the ELBO (Burda et al., 2016; Mnih & Rezende, 2016) or more expressive variational posteriors (Rezende & Mohamed, 2015; Mescheder et al., 2017). Cremer et al. (2018) provide an analysis of this gap, splitting it in two. The *approximation gap* is caused by restricting the approximate posterior to a specific parametric form, the variational family. The *amortisation gap* comes from the inference network failing to produce the optimal parameters within the family.

In this work, we address a previously unstudied aspect of amortised posterior approximations: successful approximate posterior design is not merely about picking a parametric form of a probability density. It is also about carefully deciding which conditions to feed into the inference network. This is a particular problem in sequential LVMs, where it is common practice to feed only a restricted set of inputs into the inference network: not conditioning on latent variables from other time steps can vastly improve efficiency (Bayer & Osendorfer, 2014; Lai et al., 2019). Further, leaving out

---

[*]Equal contribution.

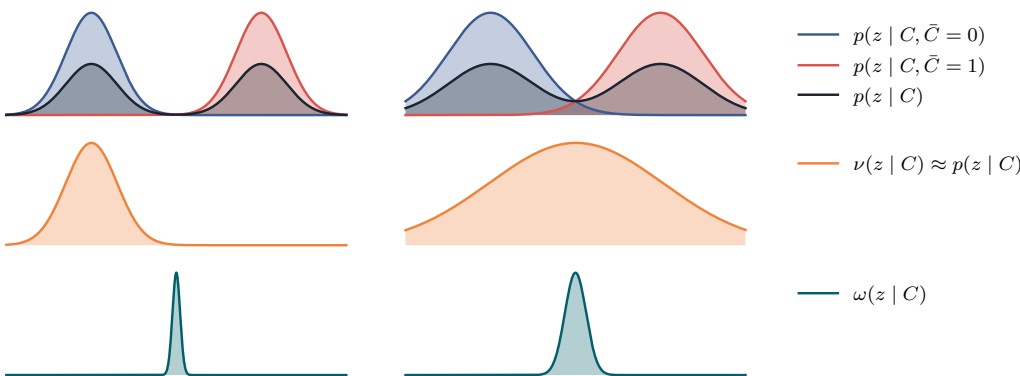

Figure 1: Illustration of the effect of partial conditioning. Consider a latent-variable model $p(C, \overline{C} \mid z) \times p(z)$, with binary $\overline{C}$ and arbitrary $C$. We omit $\overline{C}$ from the *amortised* approximate posterior $q(\mathbf{z} \mid C)$. Shown are two cases of separated (left) and overlapping (right) Gaussian true posteriors. **Top row**: the full posteriors $p(z \mid C, \overline{C} = 0)$ and $p(z \mid C, \overline{C} = 1)$ as well as their average, the marginal posterior $p(z \mid C)$. **Middle row**: Variational Gaussian approximation $\nu(\mathbf{z} \mid C)$ to the marginal posterior, which was obtained by stochastic gradient descent on the reverse, mode-seeking KL-divergence (Hoffman et al., 2013). **Bottom row**: The optimal $\omega(z \mid C)$ obtained by optimising the ELBO with a partially-conditioned amortised approximate posterior w. r. t. $q$. It is located far away from the modes, sharply peaked and shares little mass with the true full posteriors, the marginal posterior as well as the approximate marginal posterior.

observations from the future structurally mimics the Bayesian filter, useful for state estimation in the control of partially observable systems (Karl et al., 2017b; Hafner et al., 2019; Lee et al., 2020).

We analyse the emerging *conditioning gap* and the resulting suboptimality for sequential amortised inference models. If the variational posterior is only partially conditioned, all those posteriors equal w. r. t. included conditions but different on the excluded ones will need to share an approximate posterior. The result is a suboptimal compromise over many different true posteriors which cannot be mitigated by improving variational family or inference network capacity. We empirically show its effects in an extensive study on three real-world data sets on the common use case of variational state-space models.

## 2 AMORTISED VARIATIONAL INFERENCE

### 2.1 VARIATIONAL AUTO-ENCODERS

Consider the task of fitting the parameters of a latent-variable model $p_\theta(\mathbf{x}, \mathbf{z}) = p_\theta(\mathbf{x} \mid \mathbf{z}) \times p_\theta(\mathbf{z})$ to a target distribution $\hat{p}(\mathbf{x})$ by maximum likelihood:

$$\arg\max_\theta \ \mathbb{E}_{\mathbf{x} \sim \hat{p}(\mathbf{x})}[\log p_\theta(\mathbf{x})]. \tag{1}$$

In the case of variational auto-encoders (VAEs), the prior distribution $p_\theta(\mathbf{z})$ is often simple, such as a standard Gaussian. The likelihood $p_\theta(\mathbf{x} \mid \mathbf{z})$ on the other hand is mostly represented by a deep neural network producing the likelihood parameters as a function of $\mathbf{z}$.

The log marginal likelihood $\log p_\theta(\mathbf{x}) = \log \int p_\theta(\mathbf{x}, \mathbf{z}) \, d\mathbf{z}$ contains a challenging integral within a logarithm. The maximisation of eq. (1) is thus generally intractable. Yet, for a single sample $\mathbf{x}$ the log marginal likelihood can be bounded from below by the evidence lower bound (ELBO):

$$\log p_\theta(\mathbf{x}) \geq \log p_\theta(\mathbf{x}) - \mathrm{KL}(q(\mathbf{z}) \,||\, p_\theta(\mathbf{z} \mid \mathbf{x})) \tag{2}$$
$$= \mathbb{E}_{\mathbf{z} \sim q}[\log p_\theta(\mathbf{x} \mid \mathbf{z})] - \mathrm{KL}(q(\mathbf{z}) \,||\, p_\theta(\mathbf{z})) =: -\ell(\theta, q, \mathbf{x}),$$

where a surrogate distribution $q$, the variational posterior, is introduced. The bound is tighter the better $q(\mathbf{z})$ approximates $p_\theta(\mathbf{z} \mid \mathbf{x})$. The gap vanishes with the posterior KL divergence in eq. (2).

Typically, the target distribution is given by a finite data set $\mathcal{D} = \{\mathbf{x}^{(n)}\}_{n=1}^{N}$. In classical variational inference, equally many variational posteriors $\{q^{(n)}(\mathbf{z})\}_{n=1}^{N}$ are determined by direct optimisation. For instance, each Gaussian $q^{(n)}(\mathbf{z})$ is represented by its mean $\mu^{(n)}$ and standard deviation $\sigma^{(n)}$. The set of all possible distributions of that parametric form is called the variational family $\mathcal{Q} \ni q(\mathbf{z})$. Learning the generative model parameters $\theta$ can then be performed by an empirical risk minimisation approach, where the average negative ELBO over a data set is minimised:

$$\arg \min_{\theta, \{q^{(n)}\}_{n=1}^{N}} \sum_{n=1}^{N} \ell\Big(\theta, q^{(n)}(\mathbf{z}), \mathbf{x}^{(n)}\Big).$$

The situation is different for VAEs. The parameters of the approximate posterior for a specific sample $\mathbf{x}$ are produced by another deep neural network, the inference network $q_\phi(\mathbf{z} \mid \mathbf{x})$. Instead of directly optimising for sample-wise parameters, a global set of weights $\phi$ for the inference network is found by stochastic gradient-descent of the expected negative ELBO:

$$\arg \min_{\theta, \phi} \mathbb{E}_{\mathbf{x} \sim \hat{p}(\mathbf{x})}[\ell(\theta, q_\phi(\mathbf{z} \mid \mathbf{x}), \mathbf{x})].$$

This approach tends to scale more gracefully to large data sets.

**The Sequential Case**   The posterior of a sequential latent-variable model

$$p(\mathbf{x}_{1:T}, \mathbf{z}_{1:T}) = \prod_{t=1}^{T} p(\mathbf{x}_t \mid \mathbf{x}_{1:t-1}, \mathbf{z}_{1:t}) p(\mathbf{z}_t \mid \mathbf{z}_{1:t-1}, \mathbf{x}_{1:t-1}).$$

of observations $(\mathbf{x}_1, \ldots, \mathbf{x}_T) = \mathbf{x}_{1:T}$ and latents $(\mathbf{z}_1, \ldots, \mathbf{z}_T) = \mathbf{z}_{1:T}$, with $\mathbf{z}_{1:0} = \emptyset$, factorises as

$$p(\mathbf{z}_{1:T} \mid \mathbf{x}_{1:T}) = \prod_{t=1}^{T} p(\mathbf{z}_t \mid \mathbf{z}_{1:t-1}, \mathbf{x}_{1:T}). \tag{3}$$

The posterior KL divergence is a sum of step-wise divergences (Cover & Thomas, 2006):

$$\mathrm{KL}(q(\mathbf{z}_{1:T} \mid \cdot) \mid\mid p(\mathbf{z}_{1:T} \mid \mathbf{x}_{1:T})) = \sum_{t=1}^{T} \mathbb{E}_{\mathbf{z}_{1:t-1} \sim q}[\mathrm{KL}(q(\mathbf{z}_t \mid \cdot) \mid\mid p(\mathbf{z}_t \mid \mathbf{z}_{1:t-1}, \mathbf{x}_{1:T}))] \tag{4}$$

where we have left out the conditions of $q(\mathbf{z}_{1:T} \mid \cdot) = \prod_t q(\mathbf{z}_t \mid \cdot)$ for now. This decomposition allows us to focus on the non-sequential case in our theoretical analysis, even though the phenomenon most commonly occurs with sequential LVMs.

## 2.2   THE APPROXIMATION AND AMORTISATION GAPS

The ELBO and the log marginal likelihood are separated by the posterior Kullback-Leibler divergence $\mathrm{KL}(q(\mathbf{z}) \mid\mid p_\theta(\mathbf{z} \mid \mathbf{x}))$, cf. eq. (2). If the optimal member of the variational family $q^\star \in \mathcal{Q}$ is not equal to the true posterior, or $p(\mathbf{z} \mid \mathbf{x}) \notin \mathcal{Q}$, then

$$\mathrm{KL}(q^\star(\mathbf{z}) \mid\mid p_\theta(\mathbf{z} \mid \mathbf{x})) > 0,$$

so that the ELBO is strictly lower than the log marginal likelihood—the *approximation gap*.

Using an inference network we must expect that even the optimal variational posterior within the variational family is not found in general. The gap widens:

$$\mathrm{KL}(q_\phi(\mathbf{z} \mid \mathbf{x}) \mid\mid p_\theta(\mathbf{z} \mid \mathbf{x})) \geq \mathrm{KL}(q^\star(\mathbf{z}) \mid\mid p_\theta(\mathbf{z} \mid \mathbf{x})).$$

This phenomenon was first discussed by Cremer et al. (2018). The additional gap

$$\mathrm{KL}(q_\phi(\mathbf{z} \mid \mathbf{x}) \mid\mid p_\theta(\mathbf{z} \mid \mathbf{x})) - \mathrm{KL}(q^\star(\mathbf{z}) \mid\mid p_\theta(\mathbf{z} \mid \mathbf{x})) \geq 0.$$

is called the *amortisation gap*.

The two gaps represent two distinct suboptimality cases when training VAEs. The former is caused by the choice of variational family, the latter by the success of the amortised search for a good candidate $q \in \mathcal{Q}$.

Table 1: Overview of partial conditions for sequential inference networks $q(\mathbf{z}_t \mid C_t)$ in the literature. $\overline{C}_t$ denotes missing conditions vs. the true posterior according to the respective graphical model. DKF acknowledges the true factorization, but does not use $\mathbf{z}_{t-1}$ in any experiments.

| Model | $C_t$ | $\overline{C}_t$ |
|---|---|---|
| STORN (Bayer & Osendorfer, 2014) | $\mathbf{x}_{1:T}$ | $\mathbf{z}_{1:t-1}$ |
| VRNN (Chung et al., 2015) | $\mathbf{x}_{1:t}, \mathbf{z}_{1:t-1}$ | $\mathbf{x}_{t+1:T}$ |
| DKF (Krishnan et al., 2015) | $\mathbf{x}_{1:T}$ | $\mathbf{z}_{t-1}$ (exper.) |
| DKS (Krishnan et al., 2017) | $\mathbf{z}_{t-1}, \mathbf{x}_{t:T}$ | – |
| DVBF (Karl et al., 2017a) | $\mathbf{x}_t, \mathbf{z}_{t-1}$ | $\mathbf{x}_{t+1:T}$ |
| Planet (Hafner et al., 2019) | $\mathbf{x}_t, \mathbf{z}_{t-1}$ | $\mathbf{x}_{t+1:T}$ |
| SLAC (Lee et al., 2020) | $\mathbf{x}_t, \mathbf{z}_{t-1}$ | $\mathbf{x}_{t+1:T}$ |

## 3 PARTIAL CONDITIONING OF INFERENCE NETWORKS

VAEs add an additional consideration to approximate posterior design compared to classical variational inference: the choice of inputs to the inference networks. Instead of feeding the entire observation $\mathbf{x}$, one can choose to feed a strict subset $C \subset \mathbf{x}$. This is common practice in many sequential variants of the VAE, cf. table 1, typically motivated by efficiency or downstream applications. The discrepancy between inference models inputs and the true posterior conditions has been acknowledged before, e. g. by Krishnan et al. (2017), where they investigate different variants. Their analysis does not go beyond quantitative empirical comparisons. In the following, we show that such design choices lead to a distinct third cause of inference suboptimality, the *conditioning gap*.

### 3.1 MINIMISING EXPECTED KL DIVERGENCES YIELDS PRODUCTS OF POSTERIORS

The root cause is that all observations $\mathbf{x}$ that share the same included conditions $C$ but differ in the excluded conditions must now share the same approximate posterior by design. This shared approximate posterior optimizes the *expected* KL divergence

$$\omega := \arg\min_q \mathbb{E}_{\overline{C}|C}\big[\mathrm{KL}\big(q_\phi(\mathbf{z} \mid C) \,\big|\big|\, p(\mathbf{z} \mid C, \overline{C})\big)\big]. \tag{5}$$

w. r. t. the missing conditions $\overline{C} = \mathbf{x} \setminus C$. By rearranging (cf. appendix A.1 for details)

$$\mathbb{E}_{\overline{C}|C}\big[\mathrm{KL}\big(q_\phi(\mathbf{z} \mid C) \,\big|\big|\, p(\mathbf{z} \mid C, \overline{C})\big)\big]$$
$$= \mathrm{KL}\Big(q_\phi(\mathbf{z} \mid C) \,\Big|\Big|\, \exp\Big(\mathbb{E}_{\overline{C}|C}\big[\log p(\mathbf{z} \mid C, \overline{C})\big]\Big)/\mathcal{Z}\Big) - \log \mathcal{Z},$$

where $\mathcal{Z}$ denotes a normalising constant, we see that the shared optimal approximate posterior is

$$\omega(\mathbf{z}) \propto \exp\Big(\mathbb{E}_{\overline{C}|C}\big[\log p(\mathbf{z} \mid C, \overline{C})\big]\Big).$$

Interestingly, this expression bears superficial similarity with the true partially-conditioned posterior

$$p(\mathbf{z} \mid C) = \mathbb{E}_{\overline{C}|C}\big[p(\mathbf{z} \mid C, \overline{C})\big] = \exp\Big(\log\Big(\mathbb{E}_{\overline{C}|C}\big[p(\mathbf{z} \mid C, \overline{C})\big]\Big)\Big).$$

To understand the difference, consider a uniform discrete $\overline{C} \mid C$ for the sake of the argument. In this case, the expectation is an average over all missing conditions. The true posterior $p(\mathbf{z} \mid C)$ is then a mixture distribution of all plausible full posteriors $p(\mathbf{z} \mid C, \overline{C})$. The optimal approximate posterior, because of the logarithm inside the sum, is a *product* of plausible full posteriors. Such distributions occur e. g. in products of experts (Hinton, 2002) or Gaussian sensor fusion (Murphy, 2012). Products behave differently from mixtures: an intuition due to Welling (2007) is that a factor in a product can single-handedly "veto" a sample, while each term in a mixture can only "pass" it.

This intuition is highlighted in fig. 1. We can see that $\omega$ is located between the modes and sharply peaked. It shares almost no mass with both the posteriors and the marginal posterior. The approximate marginal posterior $\nu$ on the other hand either covers one or two modes, depending on the width of the two full posteriors-a much more reasonable approximation.

In fact, $\omega(\mathbf{z})$ will only coincide with either $p(\mathbf{z} \mid C)$ or $p(\mathbf{z} \mid C, \overline{C})$ in the the extreme case when $p(\mathbf{z} \mid C) = p(\mathbf{z} \mid C, \overline{C}) \Leftrightarrow p(\mathbf{z} \mid C)p(\overline{C} \mid C) = p(\mathbf{z}, \overline{C} \mid C)$, cf. appendix A.2, which implies statistical independence of $\overline{C}$ and $\mathbf{z}$ given $C$.

Notice how the suboptimality of $\omega$ assumed neither a particular variational family, nor imperfect amortisation—$\omega$ is the analytically optimal shared posterior. Both approximation and amortisation gap would only additionally affect the KL divergence inside the expectation in eq. (5). As such, the inference suboptimality described here can occur even if those vanish. We call

$$\mathbb{E}_C \left[ \mathbb{E}_{\overline{C} \mid C} \left[ \mathrm{KL}\big(\omega(\mathbf{z} \mid C) \,||\, p(\mathbf{z} \mid C, \overline{C}))\big) \right] \right] \tag{6}$$

the *conditioning gap*, which in contrast to approximation and amortisation gap can only be defined as an expectation w. r. t. $p(\mathbf{x}) = p(\overline{C}, C)$.

Notice further that the effects described here assumed the KL divergence, a natural choice due to its duality with the ELBO (cf. eq. (2)). To what extent alternative divergences (Ranganath et al., 2014; Li & Turner, 2016) exhibit similar or more favourable behaviour is left to future research.

**Learning Generative Models with Partially-Conditioned Posteriors** We find that the optimal partially-conditioned posterior will not correspond to desirable posteriors such as $p(\mathbf{z} \mid C)$ or $p(\mathbf{z} \mid C, \overline{C})$, even assuming that the variational family could represent them. This also affects learning the generative model. A simple variational calculus argument (appendix A.3) reveals that, when trained with partially-conditioned approximate posteriors, maximum-likelihood models and ELBO-optimal models are generally not the same. In fact, they coincide only in the restricted cases where $p(\mathbf{z} \mid C, \overline{C}) = q(\mathbf{z} \mid C)$. As seen before this is the case if and only if $\overline{C} \perp \mathbf{z} \mid C$.

## 3.2 LEARNING A UNIVARIATE GAUSSIAN

It is worth highlighting the results of this section in a minimal scalar linear Gaussian example. The target distribution is $\hat{p}(x) \sim \mathcal{N}(0, 1)$. We assume the latent variable model

$$p_a(x, z) = \mathcal{N}(x \mid az, 0.1) \cdot \mathcal{N}(z \mid 0, 1)$$

with free parameter $a \geq 0$. This implies

$$p_a(x) = \mathcal{N}\big(x \mid 0, 0.1 + a^2\big), \quad p_a(z \mid x) = \mathcal{N}\Big(z \mid a\big(0.1 + a^2\big)^{-1}x, \big(1 + 10a^2\big)^{-1}\Big).$$

With $a^* = \sqrt{0.9}$, we recover the target distribution with posterior $p_{a^*}(z \mid x) = \mathcal{N}(z \mid \sqrt{0.9}x, 0.1)$. Next, we introduce the variational approximation $q(z) = \mathcal{N}(z \mid \mu_z, \sigma_z^2)$. With the only condition $x$ missing, this is a deliberately extreme case of partial conditioning where $C = \emptyset$ and $\overline{C} = \{x\}$. Notice that the true posterior is a member of the variational family, i. e., no approximation gap. We maximise the *expected* ELBO

$$\omega_a(z) = \arg\max_q \mathbb{E}_{x \sim \tilde{p}} \left[ \mathbb{E}_{z \sim q} \left[ \log \frac{p_a(x, z)}{q(z)} \right] \right],$$

i. e., all observations from $\hat{p}$ share the same approximation $q$. One can show that

$$\omega_a(z) = \mathcal{N}\Big(z \mid 0, \big(100a^2 + 1\big)^{-1}\Big).$$

We immediately see that $\omega_a(z)$ is neither equal to $p_{a^*}(z \mid x) \equiv p(z \mid C, \overline{C})$ nor to $p(z) \equiv p(z \mid C)$. Less obvious is that the variance of the optimal solution is much smaller than that of both $p(z \mid C, \overline{C})$ and $p(z \mid C)$. This is a consequence of the product of the *expert compromise* discussed earlier: the (renormalised) product of Gaussian densities will always have lower variance than either of the factors. This simple example highlights how poor shared posterior approximations can become.

Further, inserting $\omega_a$ back into the expected ELBO and optimising for $a$ reveals that the maximum likelihood model parameter $a^*$ is not optimal. In other words, $p_{a^*}(x, z)$ does not optimise the expected ELBO in $p$—despite being the maximum likelihood model.

### 3.3 Extension to the Sequential Case

The conditioning gap may arise in each of the terms of eq. (4) if $q$ is partially conditioned. Yet, it is common to leave out future observations, e. g. $q(\mathbf{z}_t \mid \mathbf{z}_{1:t-1}, \mathbf{x}_{1:t}, \cancel{\mathbf{x}_{t+1:T}})$. To the best of our knowledge, in the literature only sequential applications of VAEs are potentially affected by the conditioning gap. Still, sequential latent-variable models with amortised under-conditioned posteriors have been applied success fully to, e. g., density estimation, anomaly detection, and sequential decision making (Bayer & Osendorfer, 2014; Soelch et al., 2016; Hafner et al., 2019). This may seem at odds with the previous results. The gap is not severe in all cases. Let us emphasize two "safe" cases where the gap vanishes because $\overline{C}_t \perp \mathbf{z}_t \mid C_t$ is approximately true. First, where the partially- and the fully-conditioned posterior correspond to the prior transition, i. e.

$$p(\mathbf{z}_t \mid \mathbf{z}_{t-1}) \approx p(\mathbf{z}_t \mid C_t) \approx p(\mathbf{z}_t \mid C_t, \overline{C}_t).$$

This is for example the case for deterministic dynamics where the transition is a single point mass. Second, the case where the observations are sufficient to explain the latent state, i. e.

$$p(\mathbf{z}_t \mid \mathbf{x}_t) \approx p(\mathbf{z}_t \mid C_t) \approx p(\mathbf{z}_t \mid C_t, \overline{C}_t).$$

A common case are systems with perfect state information.

Several studies have shown that performance gains of fully-conditioned over partially-conditioned approaches are negligible (Fraccaro et al., 2016; Maddison et al., 2017; Buesing et al., 2018). We conjecture that the mentioned "safe" cases are overrepresented in the studied data sets. For example, environments for reinforcement learning such as the gym or MuJoCo environments (Todorov et al., 2012; Brockman et al., 2016) feature deterministic dynamics. We will address a broader variety of cases in section 5.

## 4 Related Work

The bias of the ELBO has been studied numerous times (Nowozin, 2018; Huang & Courville, 2019). A remarkable study by Turner & Sahani (2011) contains a series of carefully designed experiments showing how different forms of assumptions on the approximate posterior let different qualities of the solutions emerge. This study predates the introduction of amortised inference (Kingma & Welling, 2014; Rezende et al., 2014) however. Cremer et al. (2018) identify two gaps arising in the context of VAEs: the approximation gap, i. e. inference error resulting from the true posterior not being a member of variational family; and the amortisation gap, i. e. imperfect inference due to an amortised inference procedure, e. g. a single neural network call. Both gaps occur *per sample* and are independent of the problems in section 3.

Applying stochastic gradient variational Bayes (Kingma & Welling, 2014; Rezende et al., 2014) to sequential models was pioneered by Bayer & Osendorfer (2014); Fabius et al. (2015); Chung et al. (2015). The inclusion of state-space assumptions was henceforth demonstrated by Archer et al. (2015); Krishnan et al. (2015; 2017); Karl et al. (2017a); Fraccaro et al. (2016; 2017); Becker-Ehmck et al. (2019). Specific applications to various domains followed, such as video prediction (Denton & Fergus, 2018), tracking (Kosiorek et al., 2018; Hsieh et al., 2018; Akhundov et al., 2019) or simultaneous localisation and mapping (Mirchev et al., 2019). Notable performance in model-based sequential decision making has also been achieved by Gregor et al. (2019); Hafner et al. (2019); Lee et al. (2020); Karl et al. (2017b); Becker-Ehmck et al. (2020). The overall benefit of latent sequential variables was however questioned by Lai et al. (2019). The performance of sequential latent-variable models in comparison to auto-regressive models was studied, arriving at the conclusion that the added stochasticity does not increase, but even decreases performance. We want to point out that their empirical study is restricted to the setting where $\mathbf{z}_t \perp \mathbf{z}_{1:t-1} \mid \mathbf{x}_{1:T}$, see the appendix of their work. We conjecture that such assumptions lead to a collapse of the model where the latent variables merely help to explain the data local in time, i. e. intra-step correlations.

A related field is that of *missing-value imputation*. Here, tasks have increasingly been solved with probabilistic models (Ledig et al., 2017; Ivanov et al., 2019). The difference to our work is the explicit nature of the missing conditions. Missing conditions are typically directly considered by learning $p(\overline{C} \mid C)$ instead of $p(\mathbf{x})$ and appropriate changes to the loss functions.

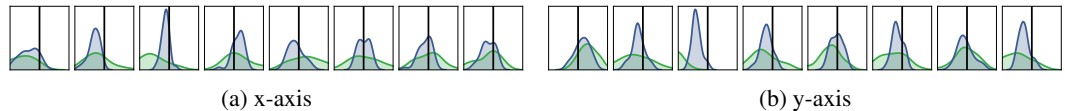

|                |                |
|:--------------:|:--------------:|
| (a) x-axis     | (b) y-axis     |

Figure 2: Posterior-predictive check of prefix-sampling on the Blackbird data set. Possible futures are sampled from the model after having observed a prefix $\mathbf{x}_{1:t}$. The state at the end of the prefix is inferred with a bootstrap particle filter. Each plot shows a kernel density estimate of the distribution over the final location $\mathbf{x}_T$, once for the semi-conditioned model in green and for the fully-conditioned in blue. The true value is marked as a vertical, black line. The fully-conditioned model assigns higher likelihood in almost all cases and is more concentrated around the truth.

## 5 STUDY: VARIATIONAL STATE-SPACE MODELS

We study the implications of section 3 for the case of variational state-space models (VSSMs). We deliberately avoid the cases of deterministic dynamics and systems with perfect state information: unmanned aerial vehicle (UAV) trajectories with imperfect state information in section 5.2, a sequential version of the MNIST data set section 5.3 and traffic flow in section 5.4. The reason is that the gap is absent in these cases, see section 3.3. In all cases, we not only report the ELBO, but conduct qualitative evaluations as well. We start out with describing the employed model in section 5.1.

### 5.1 VARIATIONAL STATE-SPACE MODELS

The two Markov assumptions that every observation only depends on the current state and every state only depends on the previous state and condition lead to the following generative model:

$$p(\mathbf{x}_{1:T}, \mathbf{z}_{1:T} \mid \mathbf{u}_{1:T}) = \prod_{t=1}^{T} p(\mathbf{x}_t \mid \mathbf{z}_t)p(\mathbf{z}_t \mid \mathbf{z}_{t-1}, \mathbf{u}_{t-1}),$$

where $\mathbf{u}_{1:T}$ are additional conditions from the data set (such as control signals) and $\mathbf{z}_0 = \mathbf{u}_0 := \emptyset$. We consider three different conditionings of the inference model

$$q(\mathbf{z}_{1:T} \mid \mathbf{x}_{1:T}, \mathbf{u}_{1:T}) = q(\mathbf{z}_1 \mid \mathbf{x}_{1:k}, \mathbf{u}_{1:k}) \prod_{t=2}^{T} q(\mathbf{z}_t \mid \mathbf{z}_{t-1}, \mathbf{x}_{1:m}, \mathbf{u}_{1:m}),$$

*partial* with $k = 1, m = t$, *semi* with $k > 1, m = t$, and *full* with $k = m = T$. We call $k$ the *sneak peek*, inspired by Karl et al. (2017a). See appendix B for more details.

### 5.2 UAV TRAJECTORIES FROM THE BLACKBIRD DATA SET

We apply semi- and fully-conditioned VSSMs to UAV modelling (Antonini et al., 2018). By discarding the rotational information, we create a system with imperfect state information, cf. section 3.3. Each observation $\mathbf{x}_t \in \mathbb{R}^3$ is the location of an unmanned aerial vehicle (UAV) in a fixed global frame. The conditions $\mathbf{u}_t \in \mathbb{R}^{14}$, consist of IMU readings, rotor speeds, and pulse-width modulation. The emission model was implemented as a Gaussian with fixed, hand-picked standard deviations, where the mean corresponds to the first three state dimensions (cf. (Akhundov et al., 2019)): $p(\mathbf{x}_t \mid \mathbf{z}_t) = \mathcal{N}\big(\boldsymbol{\mu} = \mathbf{z}_{t,1:3}, \boldsymbol{\sigma}^2 = [0.15, 0.15, 0.075]\big)$. We leave out the partially-conditioned case, as it cannot infer the higher-order derivatives necessary for rigid-body dynamics. A *sneak peek* of $k = 7$ for the semi-conditioned model is theoretically sufficient to infer those moments. See appendix C for details.

Fully-conditioned models outperform semi-conditioned ones on the test set ELBO, as can be seen in table 2a. We evaluated the models on prefix-sampling, i.e. the predictive performance of $p(\mathbf{x}_{t+1:T} \mid \mathbf{x}_{1:t}, \mathbf{u}_{1:T})$. To restrict the analysis to the found parameters of the generative model only, we inferred the filter distribution $p(\mathbf{z}_t \mid \mathbf{x}_{1:t}, \mathbf{u}_{1:t})$ using a bootstrap particle filter (Sarkka, 2013). By not using the respective approximate posteriors, we ensure fairness between the different models. Samples from the predictive distribution were obtained via ancestral sampling of the generative model. Representative samples are shown in fig. 4. We performed a posterior-predictive check for

Table 2: Results on UAV and row-wise MNIST modelling.

(a) ELBO values for models with differently conditioned variational posteriors for various data sets. Presented values are averages over ten samples from the inference model. The standard deviations were negligible. Higher is better.

|         | UAV | | Traffic Flow | |
|---------|------|------|-------|-------|
|         | val  | test | val   | test  |
| partial | -    | -    | $-2.91$ | $-2.97$ |
| semi    | 1.47 | 2.13 | $-2.73$ | $-2.75$ |
| full    | 2.03 | 2.41 | $-2.69$ | $-2.78$ |

(b) Results for row-wise MNIST. We report the ELBO as a lower bound on the log-likelihood and the KL-Divergence of the digit distribution induced by the model from a uniform distribution. Other results by Klushyn et al. (2019).

| Distribution | Log-Likelihood ↑ | KL ↓ |
|--------------|------------------|------|
| data    | -                      | 0.002 |
| partial | $\geq -98.99 \pm 0.06$ | 0.098 |
| full    | $\geq -88.45 \pm 0.05$ | 0.015 |
| vhp + rewo | $\geq -82.74$ | - |
| iwae (L=2) | $\geq -82.83$ | - |

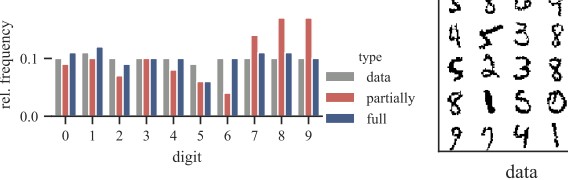 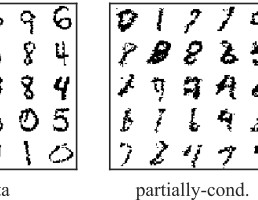 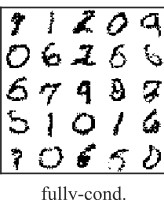

Figure 3: **Left:** Class distributions of the respective image distributions induced by a state-of-the-art classifier. The data distribution is close to uniform, except for 5. The fully-conditioned model yields too few 5's and is close to uniform for the other digits. The partially-conditioned model only captures the right frequencies of 1 and 3. **Right:** Comparison of generative sampling on row-wise MNIST. Samples from the data distribution are shown on the left. The middle and right show samples from models with an partially- and a fully-conditioned approximate posterior, respectively.

both models, where we compare the densities of the final observations $\mathbf{x}_T$ obtained from prefix sampling in fig. 2. Both evaluations qualitatively illustrate that the predictions of the fully-conditioned approach concentrate more around the true values. In particular the partially-conditioned model struggles more with long-term prediction.

## 5.3 ROW-WISE MNIST

We transformed the MNIST data set into a sequential data set by considering one row in the image plane per time step, from top to bottom. This results in stochastic dynamics: similiar initial rows can result in a 3, 8, 9 or 0, future rows are very informative (cf. section 3.3). Before all experiments, each pixel was binarised by sampling from a Bernoulli with a rate in $[0, 1]$ proportional to the corresponding pixel intensity.

The setup was identical to that of section 5.2, except that a Bernoulli likelihood parameterised by a feed-forward neural network was used: $p(\mathbf{x}_t \mid \mathbf{z}_t) = \mathcal{B}(\mathrm{FNN}_{\theta_E}(\mathbf{z}_t))$. No conditions $\mathbf{u}_{1:T}$ and a short sneak-peek ($k = 1$) were used. The fully-conditioned model outperforms the partially-conditioned by a large margin, placing it significantly closer to state-of-the-art performance, see table 2b. This is supported by samples from the model, see fig. 3. Cf. appendix D for details.

For qualitative evaluation, we used a state-of-the-art classifier[1] to categorise 10,000 samples from each of the models. If the data distribution is learned well, we expect the classifier to behave similarly on both data and generative samples, i.e. yield uniformly distributed class predictions. We report KL-divergences from the uniform distribution of the class prediction distributions in table 2b. A bar plot of the induced class distributions can be found in fig. 3. Only the fully-conditioned model is able to nearly capture a uniform distribution.

---

[1] https://github.com/keras-team/keras/blob/2.4.0/examples/mnist_cnn.py

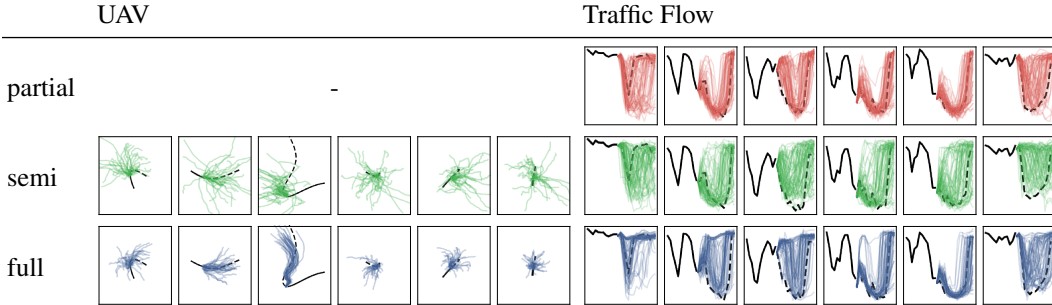

Figure 4: Comparison of prefix-sampling. Possible futures $\hat{\mathbf{x}}_{k+1:T}^{(i)}$ (coloured lines) are sampled from the model after having observed a prefix $\mathbf{x}_{1:k}$ (solid black line) and then compared to the true suffix $\mathbf{x}_{k+1:T}$ (dashed line). The state at the end of the prefix is inferred with a bootstrap particle filter. **Left:** UAV data, top-down view. **Right:** Traffic flow data, time-series view.

## 5.4 TRAFFIC FLOW

We consider the Seattle loop data set (Cui et al., 2019; 2020) of average speed measurements of cars at 323 locations on motorways near Seattle, from which we selected a single one (42). The dynamics of this system are highly stochastic, which is of special interest for our study, cf. section 3.3. Even though all days start out very similar, traffic jams can emerge suddenly. In this scenario the emission model was a feed-forward network conditioned on the whole latent state, i. e. $p(\mathbf{x}_t \mid \mathbf{z}_t) = \mathcal{N}\big(\mathrm{FNN}_{\theta_E}(\mathbf{z}_t), \sigma_x^2\big)$. We compare partially-, semi- ($k = 7$) and fully-conditioned models. See appendix E for details. The results are shown in table 2a. While the fully-conditioned posterior emerges as the best choice on the validation set, the semi-conditioned and the fully-conditioned one are on par on the test set. We suspect that the *sneak peek* is sufficient to fully capture a sensible initial state approximation.

We performed a qualitative evaluation of this scenario as well in the form of prefix sampling. Given the first $t = 12$ observations, the task is to predict the remaining ones for the day, compare section 5.2. We show the results in fig. 4. The fully-conditioned model clearly shows more concentrated yet multi-modal predictive distributions. The partially-condition model concentrates too much, and the semi-conditioned one too little.

## 6 DISCUSSION & CONCLUSION

We studied the effect of leaving out conditions of amortised posteriors, presenting strong theoretical findings and empirical evidence that it can impair maximum likelihood solutions and inference. Our work helps with approximate posterior design and provides intuitions for when full conditioning is necessary. We advocate that partially-conditioned approximate inference models should be used with caution in downstream tasks as they may not be adequate for the task, e. g., replacing a Bayesian filter. In general, we recommend conditioning inference models conforming to the true posterior.

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

## A  DETAILED DERIVATIONS

### A.1  DERIVATION OF THE SOLUTION TO AN EXPECTED KL-DIVERGENCE

We want to show that

$$
\mathbb{E}_{\overline{C}|C}\big[\mathrm{KL}\big(q_\phi(\mathbf{z} \mid C) \,\big|\big|\, p\big(\mathbf{z} \mid C, \overline{C}\big)\big)\big]
$$
$$
= \mathrm{KL}\Big(q_\phi(\mathbf{z} \mid C) \,\Big|\Big|\, \exp\Big(\mathbb{E}_{\overline{C}|C}\big[\log p\big(\mathbf{z} \mid C, \overline{C}\big)\big]\Big)/\mathcal{Z}\Big) - \log \mathcal{Z}.
$$

This can be seen from

$$
\mathbb{E}_{\overline{C}|C}\big[\mathrm{KL}\big(q_\phi(\mathbf{z} \mid C) \,\big|\big|\, p\big(\mathbf{z} \mid C, \overline{C}\big)\big)\big]
$$
$$
= \mathbb{E}_{\overline{C}|C}\bigg[\mathbb{E}_{q_\phi(\mathbf{z}|C)}\bigg[\log \frac{q_\phi(\mathbf{z} \mid C)}{p\big(\mathbf{z} \mid C, \overline{C}\big)}\bigg]\bigg]
$$
$$
= \mathbb{E}_{q_\phi(\mathbf{z}|C)}[\log q_\phi(\mathbf{z} \mid C)] - \mathbb{E}_{q_\phi(\mathbf{z}|C)}\Big[\log\Big(\exp\Big(\mathbb{E}_{\overline{C}|C}\big[\log p\big(\mathbf{z} \mid C, \overline{C}\big)\big]\Big)\Big)\Big]
$$
$$
= \mathrm{KL}\Big(q_\phi(\mathbf{z} \mid C) \,\Big|\Big|\, \exp\Big(\mathbb{E}_{\overline{C}|C}\big[\log p\big(\mathbf{z} \mid C, \overline{C}\big)\big]\Big)/\mathcal{Z}\Big) - \log \mathcal{Z},
$$

where $\mathcal{Z}$ is the normalizing constant of

$$
\exp\Big(\mathbb{E}_{\overline{C}|C}\big[\log p\big(\mathbf{z} \mid C, \overline{C}\big)\big]\Big).
$$

### A.2  ANALYZING THE OPTIMAL PARTIALLY-CONDITIONED POSTERIOR

In this section, we show that assuming either $\omega(\mathbf{z}) = p(\mathbf{z} \mid C)$ or $\omega(\mathbf{z}) = p\big(\mathbf{z} \mid C, \overline{C}\big)$ implies $p\big(\overline{C} \mid \mathbf{z}, C\big) = p\big(\overline{C} \mid C\big)$.

We begin by observing

$$
\omega(\mathbf{z}) \propto \exp\Big(\mathbb{E}_{\overline{C}|C}\big[\log p\big(\mathbf{z} \mid C, \overline{C}\big)\big]\Big)
$$
$$
= \exp\Big(\mathbb{E}_{\overline{C}|C}\Big[\log\Big(p\big(\mathbf{z} \mid C, \overline{C}\big)\frac{p(\mathbf{z} \mid C)}{p(\mathbf{z} \mid C)}\Big)\Big]\Big)
$$
$$
= p(\mathbf{z} \mid C) \exp\Big(\mathbb{E}_{\overline{C}|C}\Big[\log \frac{p\big(\mathbf{z} \mid C, \overline{C}\big)}{p(\mathbf{z} \mid C)}\Big]\Big).
$$

Now, firstly,

$$
\omega(\mathbf{z}) = p(\mathbf{z} \mid C)
$$
$$
\implies \exp\Big(\mathbb{E}_{p(\overline{C}|C)}\Big[\log \frac{p\big(\mathbf{z} \mid C, \overline{C}\big)}{p(\mathbf{z} \mid C)}\Big]\Big) = 1
$$
$$
\implies \frac{p\big(\mathbf{z} \mid C, \overline{C}\big)}{p(\mathbf{z} \mid C)} = \frac{p\big(\overline{C} \mid \mathbf{z}, C\big)}{p\big(\overline{C} \mid C\big)} = 1
$$
$$
\implies p\big(\overline{C} \mid \mathbf{z}, C\big) = p\big(\overline{C} \mid C\big).
$$

Secondly,

$$
\omega(\mathbf{z}) = p\big(\mathbf{z} \mid C, \overline{C}\big)
$$
$$
\implies \exp\Big(\mathbb{E}_{p(\overline{C}|C)}\Big[\log \frac{p\big(\mathbf{z} \mid C, \overline{C}\big)}{p(\mathbf{z} \mid C)}\Big]\Big) \propto \frac{p\big(\mathbf{z} \mid C, \overline{C}\big)}{p(\mathbf{z} \mid C)}
$$
$$
\implies \frac{p\big(\mathbf{z} \mid C, \overline{C}\big)}{p(\mathbf{z} \mid C)} = \frac{p\big(\overline{C} \mid \mathbf{z}, C\big)}{p\big(\overline{C} \mid C\big)} \text{ is constant w.\,r.\,t. } \overline{C}
$$
$$
\implies p\big(\overline{C} \mid \mathbf{z}, C\big) = p\big(\overline{C} \mid C\big).
$$

## A.3 Proof of Suboptimal Generative Model

We investigate whether a maximum likelihood solution $p^\star = \arg\min_p \mathbb{E}_{\mathbf{x}_{1:T}\sim\hat{p}}[-\log p(\mathbf{x}_{1:T})]$ is a minimum of the expected negative ELBO. From calculus of variations (Gelfand & Fomin, 2003), we derive necessary optimality conditions for maximum likelihood and the expected negative ELBO as

$$0 \overset{!}{=} \frac{\mathrm{d}\mathbb{E}_{\mathbf{x}_{1:T}\sim\hat{p}}[-\log p(\mathbf{x}_{1:T})]}{\mathrm{d}p}\alpha\frac{\mathrm{d}\mathcal{G}}{\mathrm{d}p}, \qquad \text{(max. likelihood)} \qquad (7)$$

$$0 \overset{!}{=} \frac{\mathrm{d}\mathbb{E}_{\mathbf{x}_{1:T}\sim\hat{p}}[-\log p(\mathbf{x}_{1:T})]}{\mathrm{d}p} + \frac{\mathrm{d}\mathbb{E}_{\mathbf{x}_{1:T}\sim\hat{p}}[\mathrm{KL}]}{\mathrm{d}p} + \lambda\frac{\mathrm{d}\mathcal{G}}{\mathrm{d}p}, \qquad \text{(ELBO)} \qquad (8)$$

respectively. $\mathcal{G}$ is a constraint ensuring that $p$ is a valid density, $\lambda$ and $\alpha$ are Lagrange multipliers. KL refers to the posterior divergence in eq. (2). Equating (7) and (8) and rearranging gives

$$\frac{\mathrm{d}\mathbb{E}_{\mathbf{x}_{1:T}\sim\hat{p}}[\mathrm{KL}]}{\mathrm{d}p} + (\alpha - \lambda)\frac{\mathrm{d}\mathcal{G}}{\mathrm{d}p} = 0. \qquad (9)$$

Equation (9) is a necessary and sufficient condition (van Erven & Harremoës, 2014) that the Kullback-Leibler divergence is minimised *as a function of $p$*, which happens when $p(\mathbf{z}_t \mid C_t, \overline{C}_t) = q(\mathbf{z}_t \mid C_t)$ for all $t$.

# B Model Details

We restrict our study to a class of current state-of-the-art variational sequence models: that of *residual* state-space models. The fact that inference is not performed on the full latent state, but merely the residual, allows more efficient learning through better gradient flow (Karl et al., 2017b; Fraccaro et al., 2016).

**Generative Model**  The two Markov assumptions that every observation only depends on the current state and every state only depends on the previous state and condition lead to the following model:

$$p(\mathbf{x}_{1:T}, \mathbf{z}_{1:T} \mid \mathbf{u}_{1:T}) = \prod_{t=1}^{T} p(\mathbf{x}_t \mid \mathbf{z}_t) p(\mathbf{z}_t \mid \mathbf{z}_{t-1}, \mathbf{u}_{t-1}),$$

where $\mathbf{u}_{1:T}$ are additional conditions from the data set (such as control signals) and $\mathbf{z}_0 = \mathbf{u}_0 := \emptyset$. Such models can be represented efficiently by choosing a convenient parametric form (e. g. neural networks) which map the conditions to the parameters of an appropriate distribution $\mathcal{D}$. In this work, we let the *emission model* $\mathcal{D}_{\mathbf{x}}$ be a feed-forward neural network parameterised by $\theta_E$:

$$p(\mathbf{x}_t \mid \mathbf{z}_t) = \mathcal{D}_{\mathbf{x}}(\mathrm{FNN}_{\theta_E}(\mathbf{z}_t)).$$

For example, we might use a Gaussian distribution that is parameterised by its mean and variance, which are outputs of a feed-forward neural network (FNN).

Further, we represent the *transition model* as a deterministic step plus a component-wise scaled *residual*:

$$\mathbf{z}_t = \mathrm{FNN}_{\theta_T}(\mathbf{z}_{t-1}, \mathbf{u}_{t-1}) + \mathrm{FNN}_{\theta_G}(\mathbf{z}_{t-1}) \odot \boldsymbol{\epsilon}_t, \qquad \boldsymbol{\epsilon}_t \sim \mathcal{D}_{\boldsymbol{\epsilon}}.$$

Here $\mathrm{FNN}_{\theta_G}(\mathbf{z}_{t-1})$ produces a gain with which the residual is multiplied element-wise. The residual distribution $\mathcal{D}_{\boldsymbol{\epsilon}}$ is assumed to be zero-centred and independent across time steps. Gaussian noise is hence a convenient choice.

The initial state $\mathbf{z}_1$ needs to have a separate implementation, as it has no predecessor $\mathbf{z}_0$ and is hence structurally different. In this work, we consider a base distribution that is transformed into the distribution of interest by a bijective map, ensuring tractability. We use an inverse auto-regressive flow (Kingma et al., 2016) and write $p_{\theta_{\mathbf{z}_1}}(\mathbf{z}_1)$ to indicate the dependence on trainable parameters, but do not specify the base distribution further for clarity.

**Amortised Inference Models**   This formulation allows for an efficient implementation of inference models. Given the residual, the state is a deterministic function of the preceding state—and hence it is only necessary to infer the residual. Consequently we implement the inference model as such:

$$q(\boldsymbol{\epsilon}_t \mid \mathbf{z}_{1:t-1}, \mathbf{x}_{1:T}, \mathbf{u}_{1:T}) = \hat{\mathcal{D}}_{\boldsymbol{\epsilon}_t}(\mathrm{FNN}_{\phi_{\boldsymbol{\epsilon}}}(\tilde{\mathbf{z}}_t, \mathbf{f}_t)).$$

Note that that the state-space assumptions imply $p(\boldsymbol{\epsilon}_t \mid \mathbf{z}_{1:t-1}, \mathbf{x}_{1:T}) = p(\boldsymbol{\epsilon}_t \mid \mathbf{z}_{t-1}, \mathbf{x}_{t+1:T})$ (Sarkka, 2013). We reuse the deterministic part of the transition $\tilde{\mathbf{z}}_t = \mathrm{FNN}_{\theta_T}(\mathbf{z}_{t-1}, \mathbf{u}_{t-1})$. Features $\mathbf{f}_{1:T} = \mathbf{f}_{1:T} = (\mathbf{f}_1, \mathbf{f}_2, \dots, \mathbf{f}_T)$ define whether the inference is partially-conditioned and are explained below.

Due to the absence of a predecessor, the initial latent state $\mathbf{z}_1$ needs special treatment as in the generative case. We employ a separate feed-forward neural network to yield the parameters of the variational posterior at the first time step:

$$q(\mathbf{z}_1 \mid \mathbf{x}_{1:T}, \mathbf{u}_{1:T}) = \hat{\mathcal{D}}_{\mathbf{z}_1}(\mathrm{FNN}_{\phi_{\mathbf{z}_1}}(\mathbf{f}_1)).$$

**Under-, Semi-, and Fully-Conditioned Posteriors**   We control whether a variational posterior is partially-, semi- or fully conditioned by the design of the features $\mathbf{f}_{1:T}$. In the case of a fully-conditioned approximate posterior, we employ a bidirectional RNN (Schuster & Paliwal, 1997):

$$\mathbf{f}_{1:T} = \mathrm{BiRNN}_{\phi_{\mathbf{f}_t}}(\mathbf{x}_{1:T}, \mathbf{u}_{1:T}),$$

For the case of an partially-conditioned model, a unidirectional RNN is an option. Since it is common practice (Karl et al., 2017b) to condition the initial inference model on parts of the future as well, we let the feature at the first time step *sneak peek* into a chunk of length $k$ of the future:

$$\mathbf{f}_1 = \mathrm{FNN}_{\phi_{\mathbf{f}_1}}(\mathbf{x}_{1:k}, \mathbf{u}_{1:k})$$
$$\mathbf{f}_{2:T} = \mathrm{RNN}_{\phi_{\mathbf{f}_t}}(\mathbf{x}_{1:T}, \mathbf{u}_{1:T}),$$

where the recurrent network is unidirectional. We call this approach a semi-conditioned model.

# C    DETAILS ON THE EXPERIMENTAL SETUP OF UAV MODELLING

We go with a typical split into training, validation and test data. Each consists of 10,240 sequences of 25 time steps starting out at the origin. The distribution of the residuals $\mathcal{D}_\epsilon$ and approximate posteriors $\hat{\mathcal{D}}_{\mathbf{z}_1}, \hat{\mathcal{D}}_\epsilon$ is Gaussian. A hyper-parameter search of 128 experiments was conducted. Models were trained by stochastic gradient descent using Adam (Bottou, 2010; Kingma & Ba, 2015) on the negative ELBO. We selected the best model for semi- and fully-conditioned models separately according to the respective ELBOs on the validation set after 30,000 updates, which was sufficient for all models to converge.

## C.1    HYPER-PARAMETER SEARCH SPACE

| field | range |
| --- | --- |
| batch_size | Choice([8, 16, 32, 64, 128]) |
| optimizer.learning_rate | Choice([0.003, 0.001, 0.0003, 0.0001, 3e-05]) |
| optimizer.beta1 | Choice([0.5, 0.8, 0.9]) |
| n_latent | Range(low=16, high=33) |
| initial.n_flows | Choice([2, 4, 8, 12]) |
| initial.layer_sizes.0 | Choice([8, 16, 32, 64]) |
| initial.layer_sizes.1 | Choice([8, 16, 32, 64]) |
| transition.kind | stochastic |
| transition.func_kind | Choice(['residual', 'highway', 'plain']) |
| transition.layers.0.units | Choice([16, 32, 64, 128, 192]) |
| transition.layers.0.activation | Choice(['softsign', 'softplus', 'relu', 'tanh']) |
| transition.layers.1.units | Choice([16, 32, 64, 128, 192]) |
| transition.layers.1.activation | Choice(['softsign', 'softplus', 'relu', 'tanh']) |
| inv_initial.layers.0.units | Choice([16, 32, 64, 128]) |
| inv_initial.layers.0.activation | Choice(['softsign', 'softplus', 'relu', 'tanh']) |
| inv_initial.layers.1.units | Choice([16, 32, 64, 128]) |
| inv_initial.layers.1.activation | Choice(['softsign', 'softplus', 'relu', 'tanh']) |
| inv_disturbance.layers.0.units | Choice([16, 32, 64, 128]) |
| inv_disturbance.layers.0.activation | Choice(['softsign', 'softplus', 'relu', 'tanh']) |
| inv_disturbance.layers.1.units | Choice([16, 32, 64, 128]) |
| inv_disturbance.layers.1.activation | Choice(['softsign', 'softplus', 'relu', 'tanh']) |
| feature_rnn.n_states | Choice([32, 64, 128]) |
| feature_rnn.n_layers | Choice([1, 2]) |
| feature_rnn.bidirectional | True |
| feature_rnn.cell_type | gru |
| feature_rnn.initial_mlp.prefix_length | 7 |
| feature_rnn.initial_mlp.layers.0.units | Choice([16, 32, 64, 128]) |
| feature_rnn.initial_mlp.layers.0.activation | Choice(['softsign', 'softplus', 'relu', 'tanh']) |
| feature_rnn.initial_mlp.layers.1.units | Choice([16, 32, 64, 128]) |
| feature_rnn.initial_mlp.layers.1.activation | Choice(['softsign', 'softplus', 'relu', 'tanh']) |
| emission.scale_kind | fixed |
| emission.scale | array([0.15 , 0.15 , 0.075]) |

## C.2 HYPER-PARAMETERS OF SEMI-CONDITIONED MODEL

| field | value |
| --- | --- |
| batch_size | 16 |
| optimizer.learning_rate | 0.001 |
| optimizer.beta1 | 0.9 |
| n_latent | 22 |
| initial.n_flows | 12 |
| initial.layer_sizes.0 | 8 |
| initial.layer_sizes.1 | 8 |
| transition.kind | stochastic |
| transition.func_kind | residual |
| transition.layers.0.units | 16 |
| transition.layers.0.activation | tanh |
| transition.layers.1.units | 32 |
| transition.layers.1.activation | softplus |
| inv_initial.layers.0.units | 16 |
| inv_initial.layers.0.activation | tanh |
| inv_initial.layers.1.units | 16 |
| inv_initial.layers.1.activation | softsign |
| inv_disturbance.layers.0.units | 64 |
| inv_disturbance.layers.0.activation | tanh |
| inv_disturbance.layers.1.units | 128 |
| inv_disturbance.layers.1.activation | relu |
| feature_rnn.n_states | 128 |
| feature_rnn.n_layers | 1 |
| feature_rnn.bidirectional | True |
| feature_rnn.cell_type | gru |
| feature_rnn.initial_mlp.prefix_length | 7 |
| eature_rnn.initial_mlp.layers.0.units | 32 |
| eature_rnn.initial_mlp.layers.0.activation | softplus |
| eature_rnn.initial_mlp.layers.1.units | 32 |
| eature_rnn.initial_mlp.layers.1.activation | softsign |
| emission.scale_kind | fixed |
| emission.scale | array([0.15 , 0.15 , 0.075]) |

## C.3 HYPER-PARAMETERS OF FULLY-CONDITIONED MODEL

| field | value |
| --- | --- |
| batch_size | 32 |
| optimizer.learning_rate | 0.003 |
| optimizer.beta1 | 0.8 |
| n_latent | 20 |
| initial.n_flows | 12 |
| initial.layer_sizes.0 | 16 |
| initial.layer_sizes.1 | 8 |
| transition.kind | stochastic |
| transition.func_kind | highway |
| transition.layers.0.units | 64 |
| transition.layers.0.activation | softplus |
| transition.layers.1.units | 16 |
| transition.layers.1.activation | softplus |
| inv_initial.layers.0.units | 32 |
| inv_initial.layers.0.activation | tanh |
| inv_initial.layers.1.units | 128 |
| inv_initial.layers.1.activation | relu |
| inv_disturbance.layers.0.units | 128 |
| inv_disturbance.layers.0.activation | tanh |
| inv_disturbance.layers.1.units | 128 |
| inv_disturbance.layers.1.activation | relu |
| feature_rnn.n_states | 128 |
| feature_rnn.n_layers | 1 |
| feature_rnn.bidirectional | True |
| feature_rnn.cell_type | gru |
| feature_rnn.initial_mlp.prefix_length | 7 |
| feature_rnn.initial_mlp.layers.0.units | 128 |
| feature_rnn.initial_mlp.layers.0.activation | softsign |
| feature_rnn.initial_mlp.layers.1.units | 32 |
| feature_rnn.initial_mlp.layers.1.activation | softplus |
| emission.scale_kind | fixed |
| emission.scale | array([0.15 , 0.15 , 0.075]) |

# D   DETAILS ON THE EXPERIMENTAL SETUP OF MNIST MODELLING

We conducted a hyper parameter search of 64 experiments for 15,000 iterations. The 5 best experiments (according to the ELBO at the last iteration) were continued for 85,000 further iterations. Test ELBOs are reported in table 2b.

## D.1   HYPER-PARAMETERS OF PARTIALLY-CONDITIONED MODEL

| field | range |
| --- | --- |
| batch_size | 32 |
| optimizer.learning_rate | 0.003 |
| optimizer.beta1 | 0.8 |
| n_latent | 25 |
| initial.n_flows | 12 |
| initial.layer_sizes.0 | 64 |
| initial.layer_sizes.1 | 32 |
| emission.scale_kind | learnable |
| emission.scale | 0.2 |
| emission.layers.0.units | 16 |
| emission.layers.0.activation | relu |
| emission.layers.1.units | 32 |
| emission.layers.1.activation | softplus |
| transition.kind | stochastic |
| transition.func_kind | highway |
| transition.layers.0.units | 32 |
| transition.layers.0.activation | softsign |
| transition.layers.1.units | 128 |
| transition.layers.1.activation | softplus |
| inv_initial.layers.0.units | 32 |
| inv_initial.layers.0.activation | softsign |
| inv_initial.layers.1.units | 64 |
| inv_initial.layers.1.activation | tanh |
| inv_disturbance.layers.0.units | 64 |
| inv_disturbance.layers.0.activation | softplus |
| inv_disturbance.layers.1.units | 64 |
| inv_disturbance.layers.1.activation | softsign |
| feature_rnn.n_states | 128 |
| feature_rnn.bidirectional | True |
| feature_rnn.cell_type | gru |
| feature_rnn.initial_mlp.prefix_length | 1 |
| feature_rnn.initial_mlp.layers.0.units | 16 |
| feature_rnn.initial_mlp.layers.1.units | 32 |
| feature_rnn.initial_mlp.layers.0.activation | softplus |
| feature_rnn.initial_mlp.layers.1.activation | softsign |

## D.2   HYPER-PARAMETERS OF FULLY-CONDITIONED MODEL

| field | value |
| --- | --- |
| batch_size | 128 |
| optimizer.learning_rate | 0.003 |
| optimizer.beta1 | 0.9 |
| n_latent | 25 |
| initial.n_flows | 4 |
| initial.layer_sizes.0 | 64 |
| initial.layer_sizes.1 | 32 |
| emission.scale_kind | learnable |
| emission.scale | 0.02 |
| emission.layers.0.units | 64 |
| emission.layers.0.activation | softsign |
| emission.layers.1.units | 64 |
| emission.layers.1.activation | softplus |
| transition.kind | stochastic |
| transition.func_kind | highway |
| transition.layers.0.units | 192 |
| transition.layers.0.activation | relu |
| transition.layers.1.units | 192 |
| transition.layers.1.activation | tanh |
| inv_initial.layers.0.units | 64 |
| inv_initial.layers.0.activation | softsign |
| inv_initial.layers.1.units | 16 |
| inv_initial.layers.1.activation | relu |
| inv_disturbance.layers.0.units | 64 |
| inv_disturbance.layers.0.activation | relu |
| inv_disturbance.layers.1.units | 16 |
| inv_disturbance.layers.1.activation | softsign |
| feature_rnn.n_states | 32 |
| feature_rnn.bidirectional | True |
| feature_rnn.cell_type | gru |
| feature_rnn.initial_mlp.prefix_length | 1 |
| feature_rnn.initial_mlp.layers.0.units | 16 |
| feature_rnn.initial_mlp.layers.0.activation | tanh |
| feature_rnn.initial_mlp.layers.1.units | 128 |
| feature_rnn.initial_mlp.layers.1.activation | softplus |

# E   DETAILS ON THE EXPERIMENTAL SETUP OF TRAFFIC FLOW MODELLING

The data is down-sampled to contain average speeds over 30 minute windows from 6:30 to 19:00, resulting in 26 time steps per day. The data was split into training, validation and testing data by months, January up to July for training, July to September for validation and the remainder for testing. The standard deviation $\sigma_x^2$ was determined during hyper-parameter optimisation along with the architectural and optimisation parameters. A hyper parameter search of 128 configurations was conducted. After 150,000 weight updates, the model for each partially-, semi- and fully-conditioned with the lowest negative ELBO on the validation set was selected.

## E.1   HYPER-PARAMETER SEARCH SPACE

| field | range |
|---|---|
| batch_size | Choice([8, 16, 32, 64, 128]) |
| optimizer.learning_rate | Choice([0.0003, 0.0001, 3e-05]) |
| optimizer.beta1 | Choice([0.5, 0.8, 0.9]) |
| n_latent | 64 |
| initial.n_flows | 4 |
| initial.layer_sizes.0 | Choice([8, 16, 32, 64]) |
| initial.layer_sizes.1 | Choice([8, 16, 32, 64]) |
| transition.kind | stochastic |
| transition.func_kind | Choice(['residual', 'highway', 'plain']) |
| transition.layers.0.units | 64 |
| transition.layers.0.activation | softsign |
| transition.layers.0.kernel_initia... | orthogonal |
| inv_initial.layers.0.units | 64 |
| inv_initial.layers.0.activation | softplus |
| inv_initial.layers.0.kernel_initi... | glorot_normal |
| inv_disturbance.layers.0.units | 64 |
| inv_disturbance.layers.0.activation | softplus |
| inv_disturbance.layers.0.kernel_i... | glorot_normal |
| feature_rnn.n_states | 64 |
| feature_rnn.n_layers | 1 |
| feature_rnn.bidirectional | True |
| feature_rnn.cell_type | gru |
| feature_rnn.initial_mlp.prefix_length | 7 |
| feature_rnn.initial_mlp.layers.0.units | Choice([16, 32, 64, 128]) |
| feature_rnn.initial_mlp.layers.0.activation | Choice(['softsign', 'softplus', 'relu', 'tanh']) |
| feature_rnn.initial_mlp.layers.1.units | Choice([16, 32, 64, 128]) |
| feature_rnn.initial_mlp.layers.1.activation | Choice(['softsign', 'softplus', 'relu', 'tanh']) |
| emission.scale_kind | fixed |
| emission.scale | Choice([0.1, 0.2689655172413793, 0.43793103448... |
| emission.layers.0.units | 64 |
| emission.layers.0.activation | softplus |
| emission.layers.0.kernel_initializer | glorot_normal |
| n_hidden | 64 |

## E.2 HYPER-PARAMETERS OF PARTIALLY-CONDITIONED MODEL

| field | value |
| --- | --- |
| batch_size | 8 |
| optimizer.learning_rate | 0.0001 |
| optimizer.beta1 | 0.5 |
| n_latent | 64 |
| initial.n_flows | 4 |
| initial.layer_sizes.0 | 32 |
| initial.layer_sizes.1 | 64 |
| transition.kind | stochastic |
| transition.func_kind | plain |
| transition.layers.0.units | 64 |
| transition.layers.0.activation | softsign |
| transition.layers.0.kernel_initia... | orthogonal |
| inv_initial.layers.0.units | 64 |
| inv_initial.layers.0.activation | softplus |
| inv_initial.layers.0.kernel_initi... | glorot_normal |
| inv_disturbance.layers.0.units | 64 |
| inv_disturbance.layers.0.activation | softplus |
| inv_disturbance.layers.0.kernel_i... | glorot_normal |
| feature_rnn.n_states | 64 |
| feature_rnn.n_layers | 1 |
| feature_rnn.bidirectional | True |
| feature_rnn.cell_type | gru |
| feature_rnn.initial_mlp.prefix_length | 1 |
| feature_rnn.initial_mlp.layers.0.activation | softsign |
| feature_rnn.initial_mlp.layers.0.units | 128 |
| feature_rnn.initial_mlp.layers.1.activation | relu |
| feature_rnn.initial_mlp.layers.1.units | 128 |
| emission.scale_kind | fixed |
| emission.scale | 2.97241 |
| emission.layers.0.units | 64 |
| emission.layers.0.activation | softplus |
| emission.layers.0.kernel_initializer | glorot_normal |
| n_hidden | 64 |

## E.3 HYPER-PARAMETERS OF SEMI-CONDITIONED MODEL

| field | value |
| --- | --- |
| batch_size | 128 |
| optimizer.learning_rate | 0.0003 |
| optimizer.beta1 | 0.5 |
| n_latent | 64 |
| initial.n_flows | 4 |
| initial.layer_sizes.0 | 8 |
| initial.layer_sizes.1 | 32 |
| transition.kind | stochastic |
| transition.func_kind | plain |
| transition.layers.0.units | 64 |
| transition.layers.0.activation | softsign |
| transition.layers.0.kernel_initia... | orthogonal |
| inv_initial.layers.0.units | 64 |
| inv_initial.layers.0.activation | softplus |
| inv_initial.layers.0.kernel_initi... | glorot_normal |
| inv_disturbance.layers.0.units | 64 |
| inv_disturbance.layers.0.activation | softplus |
| inv_disturbance.layers.0.kernel_i... | glorot_normal |
| feature_rnn.n_states | 64 |
| feature_rnn.n_layers | 1 |
| feature_rnn.bidirectional | True |
| feature_rnn.cell_type | gru |
| feature_rnn.initial_mlp.prefix_length | 7 |
| feature_rnn.initial_mlp.layers.0.units | 128 |
| feature_rnn.initial_mlp.layers.1.units | 16 |
| feature_rnn.initial_mlp.layers.0.activation | softsign |
| feature_rnn.initial_mlp.layers.1.activation | tanh |
| emission.scale_kind | fixed |
| emission.scale | 0.775862 |
| emission.layers.0.units | 64 |
| emission.layers.0.activation | softplus |
| emission.layers.0.kernel_initializer | glorot_normal |
| n_hidden | 64 |

## E.4  HYPER-PARAMETERS OF FULLY-CONDITIONED MODEL

| field | value |
|---|---|
| batch_size | 8 |
| optimizer.learning_rate | 0.0003 |
| optimizer.beta1 | 0.8 |
| n_latent | 64 |
| initial.n_flows | 4 |
| initial.layer_sizes.0 | 8 |
| initial.layer_sizes.1 | 8 |
| transition.kind | stochastic |
| transition.func_kind | plain |
| transition.layers.0.units | 64 |
| transition.layers.0.activation | softsign |
| transition.layers.0.kernel_initia... | orthogonal |
| inv_initial.layers.0.units | 64 |
| inv_initial.layers.0.activation | softplus |
| inv_initial.layers.0.kernel_initi... | glorot_normal |
| inv_disturbance.layers.0.units | 64 |
| inv_disturbance.layers.0.activation | softplus |
| inv_disturbance.layers.0.kernel_i... | glorot_normal |
| feature_rnn.n_states | 64 |
| feature_rnn.n_layers | 1 |
| feature_rnn.bidirectional | True |
| feature_rnn.cell_type | gru |
| feature_rnn.initial_mlp.prefix_length | 7 |
| feature_rnn.initial_mlp.layers.0.units | 64 |
| feature_rnn.initial_mlp.layers.0.activation | relu |
| feature_rnn.initial_mlp.layers.1.units | 32 |
| feature_rnn.initial_mlp.layers.1.activation | softsign |
| emission.scale_kind | fixed |
| emission.scale | 1.28276 |
| emission.layers.0.units | 64 |
| emission.layers.0.activation | softplus |
| emission.layers.0.kernel_initializer | glorot_normal |
| n_hidden | 64 |

