# OpenReview forum: "Mind the Gap when Conditioning Amortised Inference in Sequential Latent-Variable Models"
_ICLR.cc/2021/Conference — ICLR 2021 Poster_

### Official Review · AnonReviewer1 · 2020-10-23
**Good contribution**

**Rating:** 7
**Confidence:** 5

**Review:**

I enjoyed this paper, and think it provides a valuable contribution to sequental latent variable modeling of time series data.

Specifically, this paper addresses the issue of conditioning in using variational inference to fit sequential latent variable models to data. In addition to potential errors from an amortisation gap or approximation gap, a conditioning gap is identified, where a variational distribution that is not conditioned on all possible information (previous timesteps' observations and latent variables) underperforms.

This seems like an 'obvious' insight, but I think that is a strength of this paper. It clearly shows why previous work falls short of using all available information to get good performance, through a simple theoretical analysis. Further, empirically the work demonstrates how to correct for the conditioning gap.

I anticipate that through the publication of this paper at ICLR, the authors of future papers in this area will need to be careful in conditioning. This will benefit the research community as a whole, and lead to higher-quality variational approximations and papers.

One nit:

- although the optimal variational approximation may not be ideal in the theoretical study here, in Section 3.1, I think a discussion of why the KL divergence was used in this study is warranted. For example, there are other divergence measures that do not suffer from the issues presented here  (c.f. https://dataspace.princeton.edu/handle/88435/dsp01pr76f608w).  It would be helpful to point practitioners to this related work, as one option to consider given that the KL divergence learns products of posteriors and this may not be a desirable feature of a divergence for an application.

---

> ### Author Response · Authors · 2020-11-16
> **Reply**
>
> Thank you for your positive review!
>
> We will answer your specific questions here. Please also consider the general rebuttal given to all reviewers as a top-level reply to this submission’s thread.
>
> > It would be helpful to point practitioners to this related work, as one option to consider given that the KL divergence learns products of posteriors and this may not be a desirable feature of a divergence for an application.
>
> The KL is motivated by the standard VI observation that argmax_q ELBO =  argmin_q posterior KL. You raise a very interesting point here. We had not considered alternative objectives. We believe this is a promising avenue for future research. We added a short discussion to the manuscript.
> We also point you to our discussion of originality and anticipated impact in the reply to all reviews.
>
> We would love to hear how you think the paper can be improved to a clear accept in your opinion.

---

### Official Review · AnonReviewer4 · 2020-10-27
**neat and useful theoretical result supported with practical examples**

**Rating:** 7
**Confidence:** 4

**Review:**

Summary:
The paper considers the problem of Bayesian inference with partially conditioned variational posterior. Namely, this work describes the phenomena of ill-behaved variational posterior for the case of partially observed data. The paper's main theoretical finding is that the partially conditioned variational posterior behaves like a product of experts, resulting in a degenerate solution. Speaking intuitively, the true posterior can be seen as a mixture of distributions: the sum over the unobservable variable. At the same time, the optimal variational posterior mixes as a product of distributions. Clearly, the product of densities hardly depicts features of the mixture since a near-zero value of a single member is enough for zeroing out the product's density.
Nevertheless, such models are successfully applied in some cases, and the authors explain why this theoretically perspectiveless construction could work in practice. The answer is quite straightforward: the partially conditioned variational posterior works only when the unobserved variables do not affect the true posterior. Finally, the authors strengthen their theoretical studies with neat experimental studies.

Review:
It is hard to write a useful review for this paper since the authors clearly have thought through many aspects of their work. I find this paper to be a useful piece, both theoretically and practically.

My only suggestion would be to include several works into consideration. The problem of partial observability is also important for generative image models [1,2]. I don't propose to perform a model comparison, but I think the reader would benefit if you could relate your result with similar works from CV. For instance, why other models avoid this degenerate solution while still conditioning partially. Or how one can possibly escape difficulties when full conditioning is not possible on the test stage.

Minor comments:
1. page 1, section 2.1. I think by "minimization of eq. 1" the authors mean maximization of the marginal likelihood.

References:
1. Ivanov, Oleg, Michael Figurnov, and Dmitry Vetrov. "Variational autoencoder with arbitrary conditioning." arXiv preprint arXiv:1806.02382 (2018).
2. Ledig, Christian, Lucas Theis, Ferenc Huszár, Jose Caballero, Andrew Cunningham, Alejandro Acosta, Andrew Aitken et al. "Photo-realistic single image super-resolution using a generative adversarial network." In Proceedings of the IEEE conference on computer vision and pattern recognition, pp. 4681-4690. 2017.

---

> ### Author Response · Authors · 2020-11-16
> **Reply**
>
> Thank you for your positive review. As per your suggestion, we now added a discussion w.r.t. the related work you pointed out. Beyond that, we would much appreciate detailed feedback as to the necessary improvements that would merit a "clear accept" rather than a "good paper" review.
> Please also consider the general rebuttal given to all reviewers as a top-level reply to this submission’s thread.

---

> > ### Comment · AnonReviewer4 · 2020-11-21
> > **response to the authors**
> >
> > Thank you for your interesting paper.
> > I have read the rebuttal and other reviews. I still find the paper to be both novel and important for the ICLR community. Although I'm not an expert in sequential latent-variables models, the authors clearly position their work by providing references and describing relevant models. Therefore, I do not share the concerns regarding its originality.
> >
> > Answering the authors' question regarding the score raise. I sincerely believe that this submission should be accepted, and I'm inclined to raise the score in case it will affect the decision.

---

### Official Review · AnonReviewer2 · 2020-10-27

**Rating:** 7
**Confidence:** 4

**Review:**

**Summary**
This paper investigates the effect of partial conditioning on amortized inference in variational auto-encoders, focusing specifically on sequential data sources where it is common practice to have a posterior that is factorized in such a way that conditioning is partial (usually only conditioning on past signals in the sequence). Given a true posterior that is conditioned on the entire observed datapoint, the authors discuss the effect of having an approximate posterior that is only conditioned on part of the input. As the approximate posterior cannot adapt to the part of the input that is left out of the conditioning, the evidence lower bound becomes less tight, due to the larger KL divergence between the approximate posterior and the true posterior. The authors compare this to the work by Cramer et al. [1], where the distinction was made between having a restricted family of possible distributions for the approximate posterior (approximation gap) and the gap between an amortized approximate posterior with an inference network shared for all datapoints and a non-amortized approximate posterior that is optimized for each datapoint separately (amortisation gap). They argue that partial conditioning leads to a third type of gap which is distinct of the aforementioned inference gaps. Through an example with discrete observations the authors derive that when the true posterior is conditioned on the full data, and the approximate posterior is only partially conditioned, the optimal approximate posterior is something akin to a product of true posteriors over the unconditioned information, and not a mixture where the left out information is marginalized out. Through a 1D example they show that this could lead to overly sharp posteriors that have high densities in regions where the true posterior has very low density.
As the authors also state, several studies have shown that full conditioning on future observations results in negligible performance gains. However, the authors conjecture that this is because those results were mainly found on problems where the conditioning issue was not (or less) relevant.
The authors demonstrate potential performance gains on 3 datasets where conditioning on future information could be helpful: unmanned aerial vehicle trajectories from the Blackbird dataset, a sequential version of MNIST where the rows of a picture correspond to sequential observations, and a selection of a traffic flow dataset. They perform log likelihood estimates and prefix-sampling to determine the effect of conditioning (partially or fully) on future observations.


**Pros**
- The idea behind the effect of conditioning on the amortization procedure is clearly explained.
- The exposition that explains that the optimal approximate posterior could correspond to something akin to a product of distributions instead of a mixture is interesting.
- On the datasets that were selected by the authors, the benefit of full conditioning versus partial conditioning is visible in the quantitative results (log likelihood estimates).

**Cons**
- The authors argue that the conditioning gap is a distinct gap from the amortization gap that was discussed by Cramer et al. [1]. It is not clear to me why these two gaps are distinct/independent, I would say the conditioning gap is part of the amortization gap introduced by Cramer et al. since the way that conditioning is handled in amortized inference is essential to the gap between the amortized and non-amortized approximate posterior. For instance, in the example of the univariate gaussian in section 3.2, where would the amortization gap from Cramer et al. fit in as a separate gap? The amortization gap can be large because the conditioning is incomplete, or because of the limited flexibility of the neural network mapping from conditioning to parameters. Such a limited flexibility could reach the same type of error as partial conditioning.
- The effect of the narrow posteriors for partially conditioned approximate posteriors due to it being a product of distributions and not a mixture is not clear in the experiments, even though the authors do hint that this is observed in the qualitative prefix sampling experiments. The sample prefix experiments are furthermore very hard to judge, especially for the traffic flow dataset. The authors draw conclusions from these plots that I can’t confirm by looking at the plots. For instance, with respect to the traffic flow samples the authors state that “the partially conditioned model concentrates too much…”. It seems to me like it concentrates about the same amount as the full model, and I don't see how it’s “too much” as the dashed line is usually among the predictions. I think the authors are incentivized to find this conclusion because they try to argue in figure 1 that products of distributions concentrate too much argument.
- As the authors state, previous work showed that on popular datasets conditioning on future information has little gains. Even though the authors find datasets where gains can be made, these datasets are not incredibly convincing that this is actually a widespread problem for sequential VAEs. The lack of overlap between datasets that related work is evaluated on  (such as the gym or mujoco datasets or natural speech waveform datasets and polyphonic music datasets) and datasets that this work is evaluated on and the fact that the datasets of this paper are particularly small or artificial makes me a little concerned that the problems need to be cherry picked for the proposed conditioning to have an actual effect. For instance, although the MNIST example is an obvious example where conditioning on future information could help, it is fairly artificial. The authors argue that the gym and mujoco datasets have deterministic dynamics (and therefore shouldn’t suffer from partial conditioning), but do not explain why waveform datasets or polyphonic music datasets are not suitable to study this problem. Together with the fact that related work is discussed but not compared against empirically, this makes it hard to place this work in context with related work and to judge its relevance.
- I would expect more results on the influence of the sneak-peak parameter k. On the traffic flow dataset the authors suggest that the full model (with largest possible k) can perform worse on the test set than a model with intermediate k because the intermediate-k model already contains sufficient future information. This could be investigated if results were compared for models with more values of k, and a leveling off of performance gains for increasing k could confirm this conjecture.


**Minor comments**
- In section 5 there is a lot of referring to section 5 itself in the middle of sentences, which breaks the flow and seems unnecessary. See for instance the first paragraph of section 5.
- Are you using statically binarized mnist or dynamically binarized mnist?


[1] Cramer et al. inference suboptimality in variational autoencoders. https://arxiv.org/abs/1801.03558

---

> ### Author Response · Authors · 2020-11-16
> **Individual Response**
>
> Thank you for your detailed and constructive review. We are happy that you found our exposition clear and interesting, and that you consider the experiments to back up our findings quantitatively.
>
> We will answer your specific questions here. Please also consider the general rebuttal given to all reviewers as a top-level reply to this submission’s thread.
>
> > It is not clear to me why these two gaps are distinct/independent
>
> The short answer is that the approximation gap is defined on a per-sample basis, while the conditioning gap is derived for all samples, see eq. (6) in section 3.1 for the definition. For a single sample, the conditioning gap does not exist. It only emerges as soon as inference has to compromise over many samples. The effect of that compromise on the ELBO is the conditioning gap.
>
> As you correctly say, the learning can be hampered by errors of the inference network (amortisation gap) or missing inputs (conditioning gap). From this practical lens, both could be viewed as two sides of the same medal. This is a wide definition of the amortisation gap encompassing everything that makes the inference network miss q*.
>
> We cannot arrive at an equation such as “amortisation gap = conditioning gap + something”, because the LHS is per-sample and the RHS involves an expectation over all samples. This would require a redefinition of the amortisation gap.
>
> The amortisation gap measures how much the neural net deviates from the mathematically optimal solution. The conditioning gap is different. The problem is neither the network capacity, nor a particular sample x, it’s the foul compromise between all samples sharing the same C, but not necessarily the same ~C. The conditioning gap measures how much the shared optimal solution misses all the individual optimal solutions. Unlike with the amortisation gap, the target distribution has changed, so that even if the neural network adheres perfectly, the result is not desirable.
>
> We thus face two distinct phenomena, and it is worth studying them separately, as they require different remedies.
>
> > In the example of the univariate gaussian in section 3.2, where would the amortisation gap from Cramer et al. fit in as a separate gap?
>
> For *any x*, we can get a better q than omega. But not for *all x* of them at the same time.
> The q's are coupled if they share the same C, even though they differ in ~C. Hence, improving it for one ~C will make it worse for others. Notice that there is no amortisation gap here, because we can write down the solution in closed form, that is w_a(z).
> > The authors draw conclusions from these plots that I can’t confirm by looking at the plots.
>
> To convince you otherwise, we would like to draw your attention to the figure showing the prefix sampling for traffic flow, figure 4.
>
>
> The phenomenon is most clear in the second column.
> Here the fully conditioned model supports, roughly speaking, two hypotheses:
> one of a traffic jam, where the speed drops, and,
> one without a traffic jam, where the speed stays constant.
> The semi-conditioned model does so as well, although to a lesser degree.
>
> Being able to maintain several qualitatively different hypotheses of the future is essential to stochastic models. We can clearly see that conditioning more helps with that. The other columns show–more or less–the same. We did not cherry-pick those plots.
>
> Mind also that the plots are backed up by quantitative evaluations.
>
> > problems need to be cherry picked
>
> We address this in the reply to all reviewers.
>
> > I would expect more results on the influence of the sneak-peak parameter k.
>
> We acknowledge that this could be an interesting ablation study. We did not consider it central to the contribution of the paper and thus spared it.
>
> > Are you using statically binarized mnist or dynamically binarized mnist?
>
> We binarized the data before training in a consistent matter over all experiments.

---

> > ### Comment · AnonReviewer2 · 2020-11-23
> > **reply to rebuttal**
> >
> > Thank you for the excellent rebuttal.
> >
> > My questions with regard to the comparison with the amortization gap defined by Cramer et al. have been answered properly. The same goes for the reflection of the narrow posterior in the experimental results. The choice of non-standard datasets still leaves open the question (as the authors say in their general reply) whether this paper overstates the problem or if previous papers have simply focused on datasets where these problems don't show up too much. Future research will hopefully clarify this.
> >
> > I have raised my score and would like to see this paper accepted.

---

### Official Review · AnonReviewer3 · 2020-10-28

**Rating:** 6
**Confidence:** 4

**Review:**

The paper reviews the issue of partial conditioning of the amortized posterior in sequential latent variable models, typically state-space models trained with a VAE-style loss, but where the posterior used is the filtering rather than smoothing posterior. The author show that training a model with posterior with missing information can lead to a gap in estimating both the posterior and the corresponding model. They show the benefits of using the correct posteriors in simple examples.

Overall, the paper is well written, but its originality is on the low end; a large number of papers describing state-space VAE like models make explicit that the filtering posterior is technically suboptimal compared to the smoothing one; few see benefits in actually using the smoothing posterior (note that [1] derives a valid ELBO using only a one-step smoothing update). The derivation of the shared approximate posterior is standard variational inference derivations, but it is nice to see it explicitly written, and contrasted with the optimal posterior. The paper frequently gives nice intuitions behind various facts (mixture vs product of expert and the gating effect, when is an imperfectly conditioned expert enough, etc.).

The univariate Gaussian example is a good toy problem to understand the issue at hand. The numerical examples are selected to highlight the benefits of smoothing; they are interesting but perhaps not particularly challenging or surprising, and in relatively short sequences it makes sense that peeking or smoothing would benefit over filtering.


Overall, I am still inclined to accept the paper, as it investigates more clearly and makes more explicit knowledge that is usually treated as folklore and footnotes in other papers. ˆIt does not really offer methods for making smoothing posteriors actually learn more powerful models than filtering posteriors on complex datasets, nor does it technically demonstrate that SOTA state-space models have their performance limited through the information of the posterior (note that state-space models still typically underperform autoregressive ones).


Minor:
-Table 1: The Deep Kalman Filter, in some of its instantiations, has an empty \bar{C_t}; they explicitly condition z_t on z_tm1 and x_\geq t (see equation 3). See also [2] for another state space model that considers both the smoothing and filtering posteriors (as others, they note no benefit to the smoothing posterior).

-Equation 5: Technically the left hand side should be the function w, not the particular value w(z) (note z is a bound variable on one side of the equation and bound on the other side). I understand what the authors mean, but it looks strange as it is.


[1] Gregor et. al, "Temporal Difference VAE"
[2] Buesing et al., "Learning and Querying Fast Generative Models for Reinforcement Learning"

---

> ### Author Response · Authors · 2020-11-16
> **Reply**
>
> Thank you for your positive and constructive review.
>
> You mention a performance gap between state-space and auto-regressive models. Advocating for or against any of these was beyond our scope for this submission. We believe both have their merits depending on the application. Notably, SSMs are still a wide-spread tool in the engineering disciplines. If anything, we want to understand if our findings can explain some of the performance gap, but by no means claim to close it.
>
> Regarding DKF, you are right in pointing out that eq. 3 hints at a faithful posterior approximation. Yet, all four models in section 5.2 drop at least z_t-1. We have updated our overview table accordingly. We have further added the two publications pointed out to you as related work in the appropriate parts of the paper.
>
> We further point you at our reply to all reviewers, where we clarify our contribution in relation to common knowledge in the field.

---

> > ### Comment · AnonReviewer3 · 2020-11-23
> > **Reply**
> >
> > Regarding DKF: I think it does more than 'hint'- it plainly explains how to factorize the posterior. It may not do so in the experiments but the algorithmic contribution of a work can be and often is more than what accompanying experiments suggest.
> > I think the revision is however appropriate.
> >
> > I am not sure I am convinced by your argument regarding survivorship bias and novelty. I would insist the idea itself is not novel  - it is pointed out by several authors (Fraccaro, Krishnan, and others); it should come naturally to any researcher working on VAEs and familiar with smoothing in state space models (e.g. the Baum Welch algorithm). Many of these authors  who mention the methods report finding no benefit to it. Certainly, if someone had tried the idea (as I am sure many researchers have - I have personally discussed this with several researchers) on standard datasets and found clear success, they would have reported such success.  If anything, I feel the survivorship bias hypothesis works against it.
> >
> > I would have preferred the paper to acknowledge more strongly the lineage of ideas, but more importantly, for the experiment section to be less of a 'we tried X and it works great' and more of 'let's understand in what type of datasets and with which type of model does X actually work better than Y?'. In other words, what insights should the reader ideally get when reading this paper?
> > I don't think it is as simple as 'smoothing is better than filtering'.
> >
> > I do however, agree, that the issue will vary across datasets, and some datasets will benefit more smoothing. It is possible that smoothing interfaces poorly with learned models (RNNs in particular), as RNNs tend to have limited memory, and ultimately the smoothing network will in most situations result in limited lookahead.

---

> > > ### Comment · AnonReviewer3 · 2020-11-23
> > > **addendum**
> > >
> > > I should add: I remain in favor of acceptance.

---

### Author Response · Authors · 2020-11-16
**General reply**

We would like to thank the reviewers for their thoughtful and constructive feedback. Our response will first tackle two general points raised in several reviews before briefly addressing each of your reviews individually. You can find the specific rebuttals as responses to your reviews. We also invite you to read the rebuttals given to your fellow reviewers.


**Originality and Contribution**

We extend on where we see our contribution vs. what is commonly known in the literature.

We acknowledge that others have pointed out a discrepancy between the true posterior and the approximate posteriors typically used in the literature. Similarly, the approximation gap and the amortisation gap are common knowledge. In combination, it may indeed seem somewhat obvious that partial conditioning leads to suboptimal inference.

At the same time, we are not aware that partial conditioning has been questioned anywhere in the literature to a larger degree than, e.g., the choice of variational family. In fact, model names like “Deep Kalman Filters” or “Deep Variational Bayes Filters” imply that (approximate) Bayesian filters are learned. Our work shows that this is incorrect---a stronger result than expectable inference suboptimality that is apparently not obvious. In this light, we want to push back on the notion that we were merely fleshing out the obvious.

Our contribution is to go beyond a demonstration of expectable inference suboptimality. We provide theoretical and empirical evidence as well as strong intuitions for partial conditioning. It is as important as the choice of variational family or architecture of the inference network. Neither a more expressive variational family nor a more powerful network architecture can fix the problem! We want to raise awareness of this issue among researchers and hence explicitly provide intuitions to guide future design.


**Choice of Data Sets**

Some of you have inquired about our choice of non-standard data sets, since previous work saw little benefits of smoothing variants over filtering counter parts.

The presence of a conditioning gap is a property of the system that generated the data. In section 3.3., we detail two common cases where it does not surface. There is good reason to believe that previous papers focus on such data sets, as we have argued in the paper.

Why are the cases where smoothing is only as good as filtering so common in the literature then? One reason might be that we overstate the problem and it is not that dramatic at all.  Another might just be survivorship bias: papers that show bad results are less likely to be published. Due to its tendency to focus on partially-conditioned approximate posteriors, the community has also focused on data sets where this has little effect. We study three quite diverse data sets where the problem is present, and believe each of them to be as relevant as previously studied data for benchmarking variational sequence models.


**Summary of changes**

We changed the submission in the following places:

 - We added a clarification to our contribution in the introduction, based on your feedback.
 - We added a paragraph about alternative divergences as pointed out by reviewer 1.
 - We updated the definition of the conditioning gap, eq. (6) and surrounding sentence, to be properly defined for a data set rather than a subset.
 - We added missing literature that was pointed out by several reviewers.
   - Added reference to “Variational Autoencoder with Arbitrary Conditioning” by Ivanov, Oleg and Figurnov, Michael and Vetrov, Dmitry P.
    - Added reference to “Photo-Realistic Single Image Super-Resolution Using a Generative Adversarial Network” by Ledig et al.
    - Added reference to “Learning and Querying Fast Generative Models for Reinforcement Learning” by Buesing et al.
     - Added reference to “Temporal Difference Variational Auto-Encoder” by Gregor et al.
 - We have addressed all your “minor” comments.
 - We further included minor cosmetic changes for increased readability.
 - We added comment on binarisation of MNIST.

---

### Decision · Program_Chairs · 2021-01-07
**Final Decision**

**Decision:**

Accept (Poster)

**Comment:**

The paper studies how suboptimal conditioning sets create
suboptimal variational approximations in variational inference with amortization in state space models.
While the point made about the role of the conditioning set is not a new one, the point was carried out further and
more clearly in this paper than previous works. Addressing a couple of issues would
make the paper stronger:

- Really boiling down in the experiments to know for what models/data
  the "full" approach would add value would provide concrete guidance
  to the community.


- Notation choices in the paper are rough. For example, Appendix A.2
  reads like a type mismatch since the w on the left is a function of
  z but is also equal to a function of z and C.


- Adding a more detailed description of the complement of C in the
  main text